



# Brief communication: Thwaites Glacier cavity evolution

**Suzanne L. Bevan**[1], **Adrian J. Luckman**[1], **Douglas I. Benn**[2], **Susheel Adusumilli**[3], **and Anna Crawford**[2]

[1]Swansea University, Singleton Park, Swansea SA2 8PP, UK
[2]University of St Andrews, College Gate, St Andrews KY16 9AJ, UK
[3]Scripps Institution of Oceanography, University of California San Diego, La Jolla, CA, USA

**Correspondence:** Suzanne L. Bevan (s.l.bevan@swansea.ac.uk)

**Abstract.** Between 2014 and 2017, ocean melt eroded a large cavity beneath and along the western margin of the fast-flowing core of Thwaites Glacier. Here we show that from 2017 to the end of 2020 the cavity persisted but did not expand. This behaviour, of melt concentrated at the grounding line within confined sub-shelf cavities, fits with prior observations and modelling studies. We also show that acceleration and thinning of Thwaites Glacier grounded ice continued, with an increase in speed of $400\,\mathrm{m\,a^{-1}}$ and a thinning rate of at least $1.5\,\mathrm{m\,a^{-1}}$, between 2012 and 2020.

## 1  Introduction

Much of the West Antarctic Ice Sheet (WAIS) is grounded below sea level and exposed to oceanic warming at its periphery, making it a classic example of potential marine ice-sheet instability (Hughes, 1973; Schoof, 2007; Joughin et al., 2014). Any future collapse of WAIS is likely to be driven by retreat of its two largest outlet glaciers – Pine Island Glacier and Thwaites Glacier, which together, by 2013, were discharging $258\,\mathrm{Gt\,a^{-1}}$ of ice into the Amundsen Sea embayment (ASE) (Mouginot et al., 2014). See Scambos et al. (2017), and references therein, for a thorough review of the particular importance of Thwaites Glacier within the WAIS system.

Recent observations indicate that the retreat of Thwaites Glacier is already underway with satellite and airborne altimetry showing up to $2\,\mathrm{m\,a^{-1}}$ of thinning over the lower reaches of the glacier (McMillan et al., 2014). From 2006, after 14 years of steady flow, the main trunk of Thwaites Glacier began to accelerate. By 2013, velocity accelerations from 3 to $4\,\mathrm{km\,a^{-1}}$ had led to a 33 % increase in ice flux across the grounding line (Mouginot et al., 2014).

Dynamic change was accompanied by grounding line retreat. Beneath its central fast-flowing region the grounding line of Thwaites Glacier retreated inland down a retrograde bed slope by 12–18 km between 1996 and 2011 (Rignot et al., 2014) (Fig. 1a). The grounding lines were mapped using differential interferometry applied to satellite-borne synthetic aperture radar (SAR) images from ERS-1 and ERS-2 and were updated by Milillo et al. (2019) using the same technique applied to COSMO-SkyMed data. Milillo et al. (2019) found that by 2016/17 the grounding lines in many locations had further retreated. In particular, the grounding line parallel to and west of the fast-flowing core had retreated by up to 3.2 km compared with 2011, migrating back and forth with the tidal cycle across a broad 2.5 km grounding zone.

Between 2011 and 2014, surface elevations of the ice in the area where the new grounding zone was to develop decreased by around $4\,\mathrm{m\,a^{-1}}$. From mid-2014, as the ice went afloat and began to melt from below, thinning rates based on reductions in hydrostatic thicknesses increased to $200\,\mathrm{m\,a^{-1}}$ (Milillo et al., 2019). By late 2016, a 350 m deep, $4 \times 10\,\mathrm{km}$ cavity could be identified in radar depth soundings. Whilst the initial thinning was driven dynamically, the high melt rates within the new cavity were likely sustained by the intrusion of warm modified Circumpolar Deep Water (mCDW) (Nakayama et al., 2019). The dense warm mCDW crosses the continental shelf and can access the ASE ice-shelf grounding lines via bathymetric troughs. Decadal variability in the flow of mCDW onto the continental shelf drives ASE ice-shelf thinning and glacier retreat on corresponding timescales modified by local bed geometry (Jenkins et al., 2018).

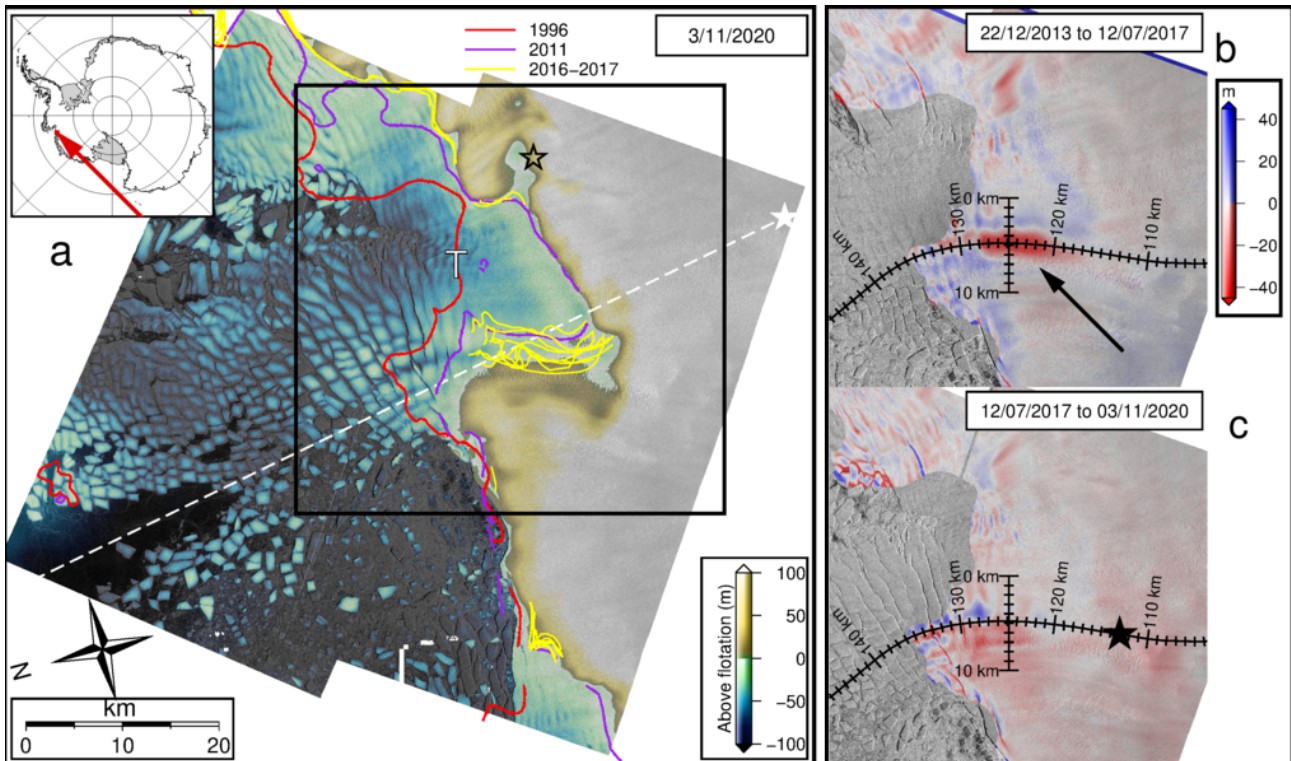

**Figure 1. (a)** TanDEM-X DEM-derived height above flotation for 3 November 2020. MEaSUREs (Rignot et al., 2016) grounding lines in red (1996) and purple (2011) and Milillo et al. (2019) grounding lines in yellow. Black and white stars mark the tie point locations for each DEM frame. The ICESat-2 track is shown by the dashed white line. The letter T shows the point used for the tide-height model. The black box shows the area covered by **(b)** and **(c)**. **(b)** Elevation change based on TanDEM-X DEMs from 22 December 2013 to 12 July 2017. Flow lines and transects in black correspond to those plotted in Fig. 2. The black arrow indicates the cavity referred to in the text. **(c)** As for **(b)** except elevation change is from 12 July 2017 to 3 November 2020. The black star marks the velocity extraction location for Fig. 3. Background shading on all panels is TerraSAR-X backscatter intensity.

In this study we use an extended time series of TanDEM-X digital elevation models (DEMs) and an updated bathymetry (Jordan et al., 2020) to examine the ongoing evolution of the new cavity. We also extend the record of wider area elevation change and trends in surface velocities using TerraSAR-X and Sentinel-1 data.

## 2  Methods

### 2.1  TanDEM-X DEMs

We created a time series of 152 DEMs from June 2011 to November 2020 based on experimental SAR data from the TanDEM-X satellite system. We used Gamma Remote Sensing software to interfere, unwrap, and phase scale (with the provided orbit vector data) the 2 m bistatic strip-map mode Co-registered Single-look Slant-range Complex images (CoSSCs). We initially geocoded the slant-range geometry DEMs to a horizontal resolution of 8 m using the RAMP DEM (Liu et al., 2015) gap-filled using the REMA DEM (Howat et al., 2019) and then iteratively refined the geocod-

ing using the interferometrically generated DEM itself. The area is covered by two satellite scenes (Fig. 1a) mostly acquired on consecutive days (Table A1). We calibrated all of the 86 southernmost DEMs in the vertical using an ICESat-2 elevation point acquired on 5 November 2018. Whilst it would have been preferable to calibrate the DEMs individually with cotemporal ICESat-2 data, ICESat-2 did not begin to acquire data until October 2018. The temporal and spatial sparsity of ICESat-2 tracks meant that only eight TanDEM-X scenes had altimeter measurements acquired within 3 d (Table A1). We therefore took the approach of using a single ICESat-2 measurement acquired at a high-elevation, slow-moving location where there was minimal likelihood of temporal elevation change (Fig. 1a). The 66 adjacent DEMs lacked a suitable high-elevation location for a tie point so we tied them to their cotemporal neighbours using a point within the scene overlap. We minimized elevation errors in the adjacent DEMs caused by the combination of topography and geolocation in this process by choosing the point over a relatively flat region. To validate this approach, Appendix Table A1 lists the mean elevation differences between scene

pairs in a $4 \times 4$ km patch above the grounding line and in the overlap region. The patch covers $500 \times 500$ pixels. The majority of the mean and standard deviations of the differences are less than 1 m. This difference will contribute to the height uncertainty in the north scenes, but the majority of the analysis to follow concentrates on surface elevations of the south scenes.

For elevations derived from ICESat-2 laser altimetry, we used data provided in the ATLAS/ICESat-2 L3A Land Ice Height (ATL06) product (Smith et al., 2020). The spatial footprint of ICESat-2 is 17 m, and we use the "atl06_quality_summary" field to remove low-quality data due to high surface slope or roughness, high uncertainty in surface height, unreliable higher-level data, or cloudy conditions (Smith et al., 2019).

In order to map floating areas, we adjusted the elevations from the WGS84 ellipsoid datum to the EGM2008 geoid (Pavlis et al., 2012) adding 1.81 m to allow for local mean dynamic topography (Armitage et al., 2018). We corrected for tide height using the Circum-Antarctic Tidal Simulation version 2008 (CATS:2008) model (Howard and Padman, 2015). As the model does not cover the full area of interest following grounding line retreat, we extracted the tide height at a point offshore (75.36° S, 106.60° W). We then assumed that any floating ice was in hydrostatic equilibrium to calculate a thickness ($H$) from the adjusted elevations ($h$) using Eq. (1) with an ice density ($\rho_i$) of 917 kg m$^{-3}$, seawater density ($\rho_w$) of 1028 kg m$^{-3}$, and firn air depth $f_a$ of 16 m.

$$H = \left( (h - f_a) \times \frac{\rho_w}{\rho_w - \rho_i} \right) + f_a \tag{1}$$

Where elevations minus hydrostatic thicknesses were above the bed depth (Jordan et al., 2020) the ice was assumed to be floating. The value for firn air depth can make a large difference to the floatation height. Using $f_a = 16$ m where the ice thickness is 800 m is equivalent to a mean ice density of 898 kg m$^{-3}$. Other studies that have attempted to correct ice density in Antarctic ice shelves either directly or via incorporating firn air depths have used values including 13–19 m of firn air (Griggs and Bamber, 2011) or 904 kg m$^{-3}$ (Khazendar et al., 2016). We chose a value of 16 m following Jordan et al. (2020), who found that this value resulted in good agreement between grounding zones inferred from the REMA DEM and interferometrically determined grounding zones (Rignot et al., 2014).

### 2.1.1 Uncertainties in surface elevations

Orbit and hence baseline uncertainties (Rizzoli et al., 2017) mean that uncertainties in relative elevation across a scene will be of the order of 1 m. Our method of vertically tieing adjacent scenes within the overlap region means that errors owing to baseline uncertainties in the south scene will be propagated into the north scene and will therefore be up to 2 m. This uncertainty will be in addition to that resulting from any mis-registration between the two scenes as described earlier. Elevation errors may also arise owing to spatially variable surface penetration depths of the X-band SAR compared with the calibration location. We used the eight TanDEM-X DEMs with near cotemporal ICESat-2 elevations to assess the accuracy of the TanDEM-X DEMs. Using only points that were measured on grounded ice, the mean TanDEM-X minus ICESat-2 difference over 23 147 point elevations in the south scene was $-0.7$ m (standard deviation 2.2 m), and over 877 points in the north scene the difference was $+2.0$ m (standard deviation 1.8 m) (see Fig. A1d). From this comparison we conclude that uncertainties based on a combination of baseline errors, surface penetration, and, for the northern scene, mis-registration are $\pm 2$ m (equivalent to $\pm 16$ m in thickness of floating ice).

In addition, any temporal elevation change at the tie point will add a time-varying bias in elevation. Elevation change rates at the tie-point location, between 2009 and 2012, from Fig. 3 in McMillan et al. (2014) appear to be about 0.5 m a$^{-1}$. As the tie-point elevation was measured in October 2018, this rate of elevation change would equate to a bias ranging from $-3.5$ to $+1.0$ m over the 9-year time span of our DEMs. In summary, we estimate elevation uncertainties to be $\pm 2$ m plus the time-varying bias.

### 2.2 Ice surface velocities

We measured ice surface velocities by feature tracking (e.g. Strozzi et al., 2002) pairs of both Sentinel-1A and B single-look complex (SLC) images (2015–2020) and TerraSAR-X SLCs (2012–2020), using Gamma Remote Sensing software. The freely available Sentinel-1 SAR data were downloaded from archives of the EU Copernicus programme. Sentinel-1A began acquiring data in 2014 with repeat coverage once every 12 d, and the launch of Sentinel-1B improved repeat coverage to once every 6 d from September 2016. We used all available 6 and 12 d pairs for feature tracking, a total of 558 pairs. The TerraSAR-X SLCs used were the master images of the TanDEM-X SLC pairs used to create the DEMs. As can be seen from Table A1, the possible delays between available images ranged from 11 d to large multiples thereof. We tracked only those pairs with delays of 44 d or less, 23 pairs in total. For the Sentinel-1 data tracking, search patch sizes were 416 range $\times$ 128 azimuth pixels, which is equivalent to about 1000 m in range and in azimuth. Tracking was carried out at a sampling of 50 range $\times$ 10 azimuth pixels, equating to approximately 100 m in ground coordinates. For the TerraSAR-X tracking, the search patch size was 128 range $\times$ 128 azimuth pixels, which is equivalent to about 30 m in range and 40 m in azimuth. Offsets were calculated every 20 pixels in range and azimuth, equivalent to approximately 40 m in ground coordinates. Following standard gamma CE1 procedures, master and slave patch sizes were equal, and cross-correlation was achieved in the spatial frequency domain. Following the tracking, offsets were con-

verted from slant to ground range coordinates. The displacements were filtered in space: first by a signal-to-noise ratio based on the cross-correlation of image patches and then again using deviation from the mean displacement within a neighbourhood. The results were geocoded to a polar stereographic projection (EPSG:3031) and the range and azimuth displacements reprojected to present them relative to grid north. Conversion to ground-range coordinates and geocoding was completed using the REMA DEM (Howat et al., 2019) gap-filled with the RAMP DEM (Liu et al., 2015).

## 3 Results

TanDEM-X DEM heights above flotation for November 2020 (Fig. 1a) show little retreat from the DInSAR-derived grounding lines from 2016/17, with the exception of the small region upstream of the cavity. This region, similar to the lobed region on the opposite eastern side of the glacier close to the black star in Fig. A1, is apparent and expands throughout the time series. In 2011 and 2017 the DEM-derived grounding lines match the DInSAR ones in the region of the cavity and near the fast-flowing central part of the glacier. To the east of the glacier the 2011 DInSAR grounding lines skirt locations that appear as disconnected pinning points (Fig. A1a) that have disappeared by 2017 in both the DInSAR and DEM (Fig. A1c). The largest discrepancies between DInSAR- and DEM-derived grounding lines are in 2011 to the west of the main glacier trunk. This discrepancy may be due to acquisition time – the ERS-2 data were from April 2011 and the TanDEM-X data are 22/23 June 2011, and this is a location of grounding line retreat.

Surface elevation losses of 50 to 60 m between 2014 and 2017 over the cavity location (Fig. 1b) confirm those presented by Milillo et al. (2019) (their Fig. S6). The ice here was inferred to have gone afloat during 2014, and basal melt rates were estimated to be up to 200 m a$^{-1}$. Elevation changes between 2017 and 2020 (Fig. 1c) show that the earlier cavity erosion process has slowed but that the cavity has persisted; in other words only small changes in thickness have taken place.

We extracted profiles of surface elevation and ice base along a flow line and a cross-flow transect. The flow line was based on the mean velocity direction and was constructed to pass through the area of maximum thickness change. Both profile locations are shown in Fig. 1b and c. The ice base was obtained by subtracting the hydrostatic thickness from the surface elevation. Where the ice is grounded the plots show the ice base below bedrock; this is simply a device to indicate height above flotation scaled by $\rho_w/(\rho_w - \rho_i) \approx 8$ (Fig. 2). We found a good agreement between the range of 2016/17 interferometric grounding lines and the locations where the inferred ice base is close to the bedrock between 115 and 118.5 km along the flow line. The flow-line profiles (Fig. 2a) show that the ungrounding evolves spatially in the up-flow

direction. As early as 2011 a small cavity apparently existed at 115 km along the flow line; we have no evidence that this cavity connected to the ocean until June 2013. A second cavity develops by April 2015, by which time we can identify a path to the ocean in a direction perpendicular to the flow. By June 2016 the cavities have merged, and in January 2017 they connect with the existing downstream ice shelf. Beyond 2017 and up to the end of 2020 the cavity remains stable. The temporal evolution can be seen more easily in the supplementary animation.

The cross-flow profiles (Fig. 2b) show the cavity expanding steadily inland from 2011 to 2019 with a good agreement to DInSAR grounding line locations in 2011 and 2016/17 and confirming the $> 2$ km grounding line migration zone after 2014. Our estimated cavity depths along both profiles are up to 200 m. Our 2016/17 DEM-based grounding lines alongside the cavity do not show the large spatial variation indicated by the DInSAR method but delineate the most retreated location (Fig. 2a), probably as the ice here is so close to flotation and the DInSAR lines migrate across a grounding zone.

Thwaites Glacier and floating tongue continue to accelerate from 2012 to 2020. Over much of the fast-flowing region speeds are 400 m a$^{-1}$ greater in January 2021 than in January 2012 – an increase of more than 10 % (Fig. A2a). Velocities at a point about 5–10 km upstream of the cavity location (Fig. 1c) increase at an average annual rate of 70 m a$^{-1}$ with steeper acceleration since mid-2015 and intra-annual variability of the order of 0.1 CE2 (Fig. 3). Observed thinning at this location is about 1.5 m a$^{-1}$, but this is likely to be an underestimate owing to the temporally changing bias in elevation uncertainty, and the thinning rate could be as great as 2.0 m a$^{-1}$.

## 4 Discussion

Our results show that the cavity beneath the newly floating region along the western border of Thwaites Glacier has not continued to deepen beyond 2017. The stability of the grounding lines, which are now in regions of prograde bed slopes, indicates that the advection of ice here is matched by high thinning rates, due to either melt or dynamic thinning. Extremely high melt rates, up to 200 m a$^{-1}$ (Milillo et al., 2019), were detected in the cavity between 2014 and 2017 and may now contribute to maintaining the new grounding line positions. However, this melt has not continued to increase the cavity depth, a fact that is consistent with observations and model studies showing that high melt rates within shallow cavities are restricted to the vicinity of the grounding line. For example, new cavities exposed since 1993 beneath ASE ice shelves remain on average just 112 m thick with 95 % of them less than 400 m deep; and the sub-shelf topography continues to closely follow the contours of the bed topography (Jordan et al., 2020).

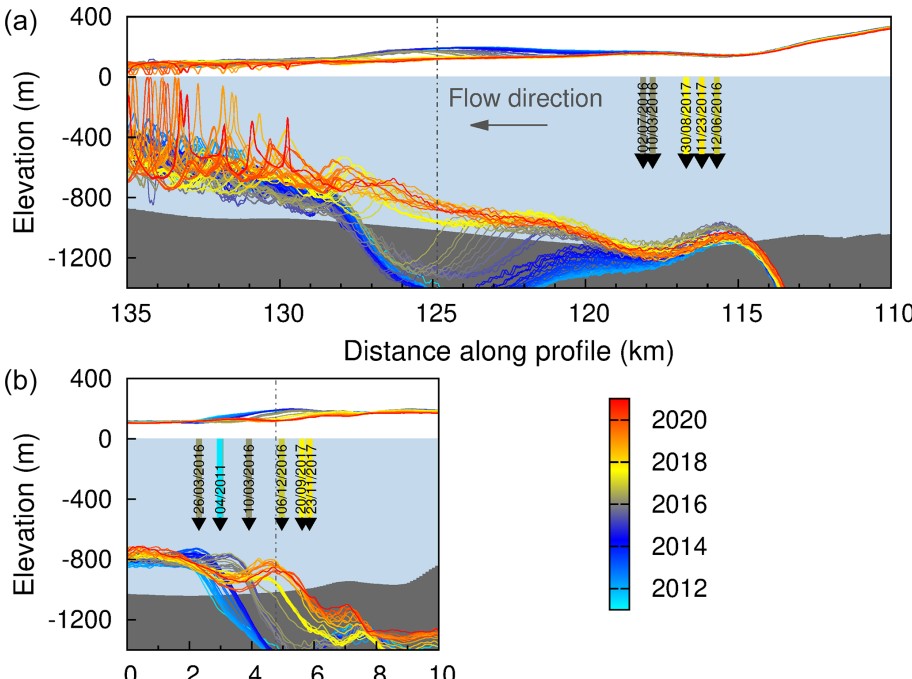

**Figure 2.** Ice surface elevation and hydrostatic thickness extracted from the flow-line profile **(a)** and the across-flow transect **(b)** marked in Fig. 1b. Where the ice base assuming hydrostatic thickness lies below the bedrock elevation this should be interpreted as a scaled height above floatation where the scaling factor is $\rho_{\rm w}/(\rho_{\rm w} - \rho_{\rm i}) \approx 8$. Vertical dashed lines mark the intersections of the two profiles. Coloured arrows indicate grounding line locations for 2011 (Rignot et al., 2016) and 2016/17 (Milillo et al., 2019) coloured according to date.

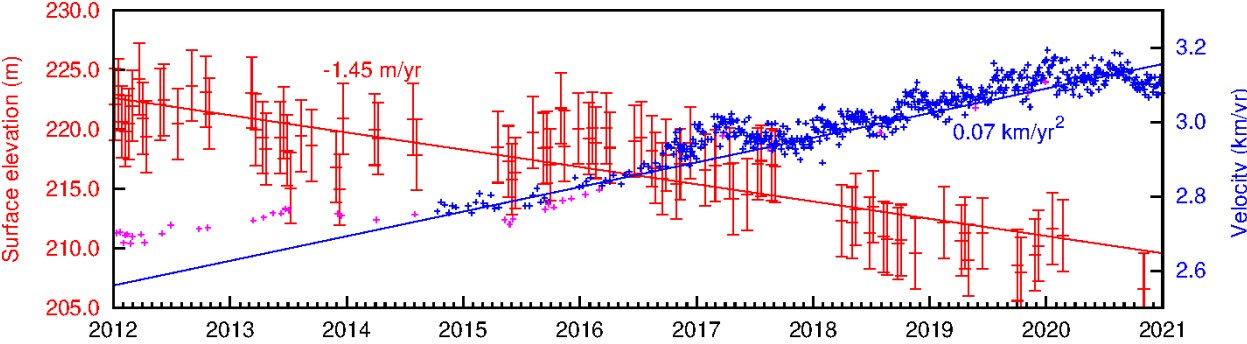

**Figure 3.** Time series of heights (relative to the EGM2008 geoid; Pavlis et al., 2012) and surface speeds extracted at the point marked by the star in Fig. 1c. Vertical bars on the elevation points represent the ±2 m estimated error. Magenta crosses represent the TerraSAR-X speeds, and blue crosses represent the Sentinel-1 speeds.

Although coupled ice–ocean models (that compare well with observed retreat rates) show that thinning rates are initially high beneath newly ungrounded ice, after 1 or 2 years the ocean circulation within the cavity adjusts, and melt and thinning become concentrated along the high-basal-slope regions close to the grounding line (Goldberg et al., 2012; Seroussi et al., 2017). On the same timescale, melt close to the grounding line increases the local basal slope until the melt rate is balanced by advection of thicker ice. Once ocean circulation and basal slope have adjusted, the cavity geometry can remain stable. Without the ocean coupling, models that parameterize melt rate using, for example, a simple depth-dependent rate tend to overestimate the delivery of ocean heat and melt near the grounding line and hence predict unrealistic grounding line retreat. The timescales of the adjustment of ocean circulation and ice-base geometry may explain why the cavity beneath Thwaites Glacier expands for a few years and then maintains its shape. The fast flow of the ice and the locally restricted melting mean that downstream cavity depth remains shallow.

We show that in 2020, steadily increasing velocities and dynamic thinning of grounded ice continue beyond estimates

published up to 2014, and although we measure thinning rates at the location plotted in Fig. 3 that are lower than the $4\,\mathrm{m\,a^{-1}}$ measured by Milillo et al. (2019) our methodology means that we may be underestimating the rate by up to $0.5\,\mathrm{m\,a^{-1}}$. Bed topography and ice thickness close to flotation can mean that the observed long-term and steady thinning of Thwaites and other WAIS glaciers in ASE can cause a rapid local thinning induced by a melt–flotation feedback, until ocean circulation adjusts. Coupled ocean–ice modelling also suggests that further ice-shelf thinning and acceleration of inland ice, where the bed slopes upwards inland, takes place over longer 10–100-year timescales (Goldberg et al., 2012) so that further consequences of the 2011–2016 grounding-line retreat, here and elsewhere on Thwaites Glacier, may not have become apparent yet.

Using InSAR DEMs to delineate grounding lines or zones is a complementary method to using DInSAR. The DEM method requires accurate surface and bed elevations, and a good estimate of ice density whereas the DInSAR method requires a detectable response of the floating ice to changing tidal elevations. Using the DEM method has allowed us to create repeated full-coverage mapping and to identify locations where there is insufficient ice-shelf flexure to allow detection by the DInSAR method but where ocean water ingress is possible and likely to result in further melt. The DEM method may miss a grounding line retreat where the grounding line is not in hydrostatic equilibrium, and retreat may even cause surface uplift. It will be important to confirm the evolution of the grounding lines with interferometric analysis.

## 5  Conclusions

Using a time series of DEMs based on interferometric processing of TanDEM-X SAR images, we have shown that the 2014–2017 grounding-line retreat and cavity development beneath the western flank of Thwaites Glacier persist with little change to the end of 2020. Based on existing model-based understanding, we conclude that restricted ocean circulation within the cavity and concentration of melt at the grounding line are responsible for the maintenance of the cavity. On a wider-scale perspective, in 2020 acceleration and dynamic thinning of Thwaites Glacier continue at a similar rate beyond that already observed up to 2014.

# Appendix A

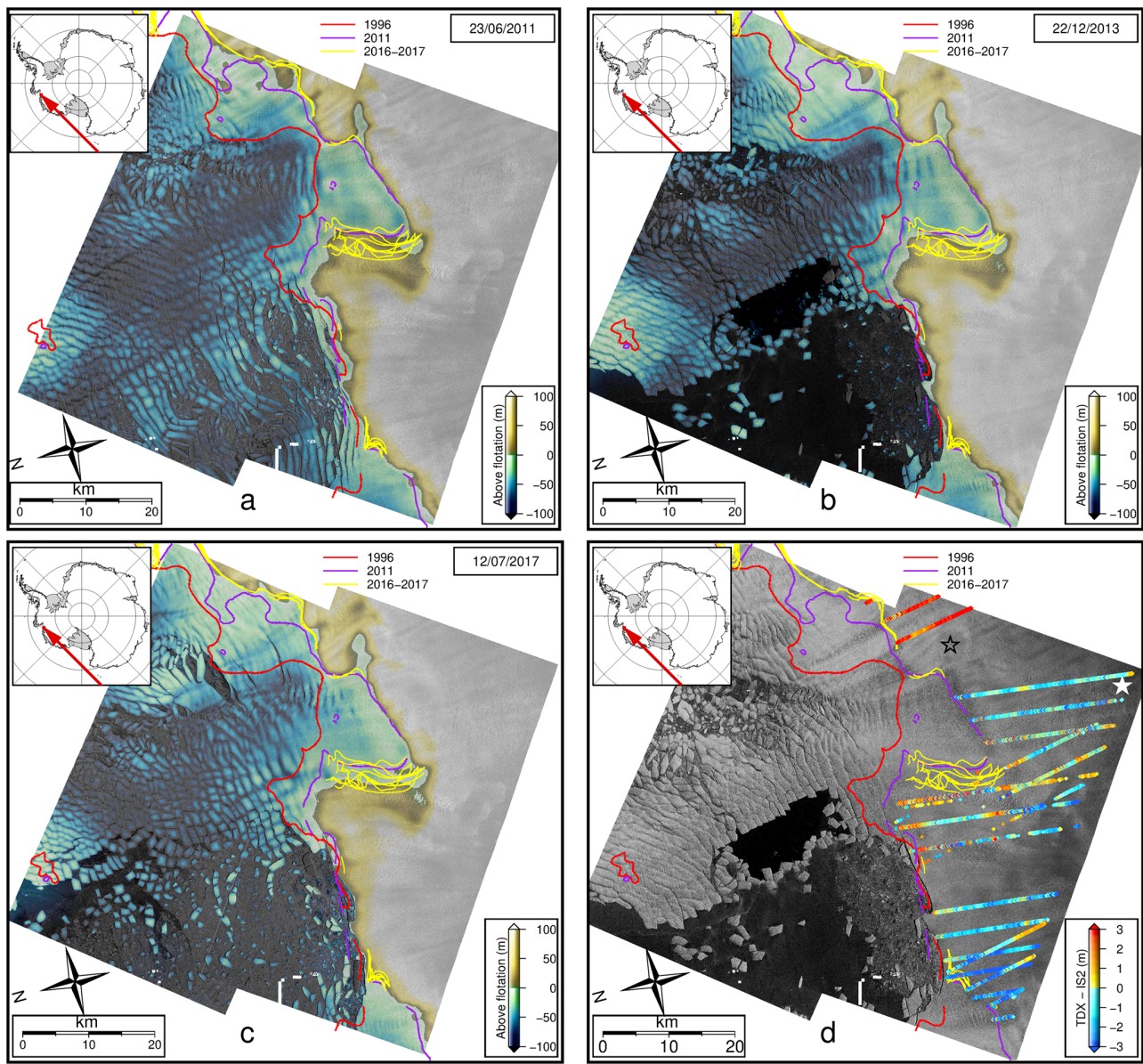

**Figure A1.** TanDEM-X DEM-derived height above flotation for **(a)** 23 June 2011, **(b)** 2 December 2013, and **(c)** 12 July 2017. MEaSUREs (Rignot et al., 2016) grounding lines in red (1996) and purple (2011) and Milillo et al. (2019) grounding lines in yellow. **(d)** TanDEM-X elevation minus ICESat-2 elevation for all points over grounded ice, when there was no more than 3 d between the TanDEM-X and ICESat-2 acquisitions. Black and white stars mark the tie point locations for each DEM frame.

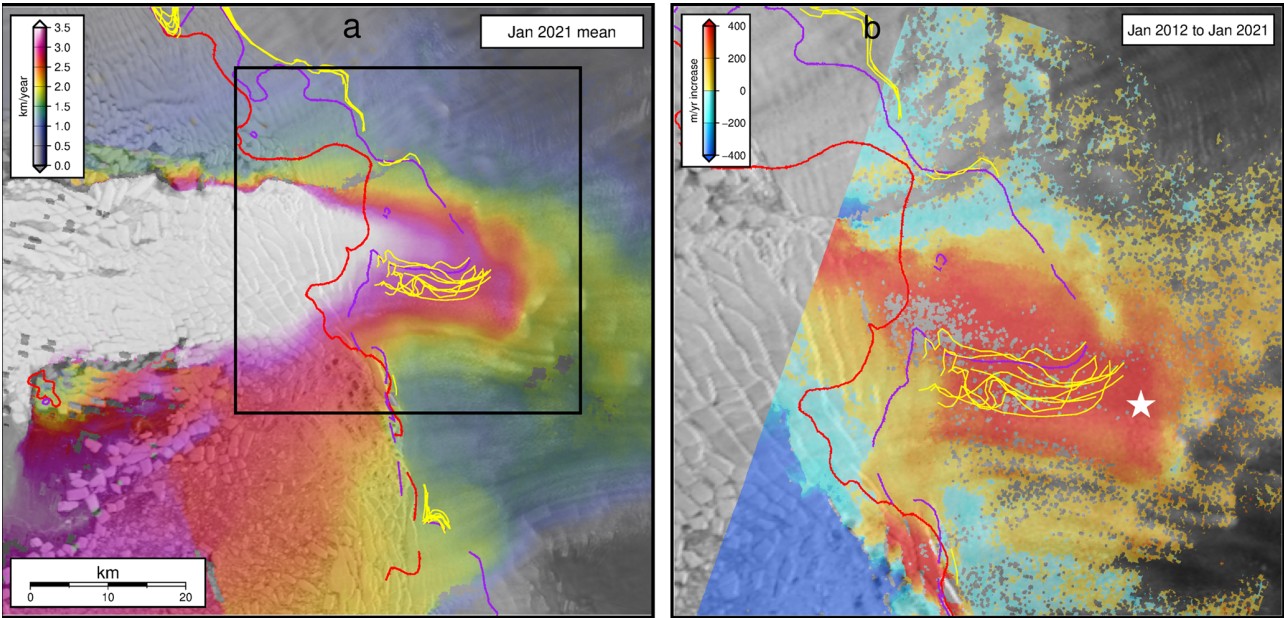

**Figure A2. (a)** Mean surface speeds for January 2021 based on feature tracking Sentinel-1 data. **(b)** Change in surface speed from January 2012 based on feature tracking TerraSAR-X data to January 2021 speeds. The white star marks the velocity extraction point.

**Table A1.** TanDEM-X DEM dates and times (UTC). Dates in bold have ICESat-2 points acquired within $\pm 3$ d. The overlap elevation differences are based on a $4 \times 4$ km ($500 \times 500$ pixel) patch centred at ($75.43°$ S, $106.31°$ W), over grounded ice.

| South scene | North scene | Overlap difference Mean (m) | SD (m) |
|---|---|---|---|
| 22 June 2011 04:39:00 | 23 June 2011 04:22:00 | 0.93 | 0.51 |
| 6 January 2012 04:39:02 | 7 January 2012 04:22:00 | 1.03 | 0.58 |
| 17 January 2012 04:39:01 | 18 January 2012 04:22:00 | 0.53 | 0.45 |
| 28 January 2012 04:39:00 | 29 January 2012 04:22:00 | −0.36 | 0.36 |
| 8 February 2012 04:39:01 | 9 February 2012 04:22:00 | −0.56 | 0.34 |
| 19 February 2012 04:39:00 | 20 February 2012 04:21:59 | −0.52 | 0.36 |
| 1 March 2012 04:39:01 | 2 March 2012 04:22:00 | −0.09 | 0.34 |
| 23 March 2012 04:39:01 | 24 March 2012 04:22:00 | 0.77 | 0.52 |
| 3 April 2012 04:39:01 | 4 April 2012 04:22:01 | 0.08 | 0.32 |
| 14 April 2012 04:39:02 | 15 April 2012 04:22:01 | 0.98 | 0.50 |
| 28 May 2012 04:39:05 | 29 May 2012 04:22:04 | 0.06 | 0.39 |
| 8 June 2012 04:39:05 | 9 June 2012 04:22:04 | 0.30 | 0.37 |
| 22 July 2012 04:39:07 | 23 July 2012 04:22:07 | −0.04 | 0.36 |
| 4 September 2012 04:39:08 | 5 September 2012 04:22:07 | 0.54 | 0.50 |
| 18 October 2012 04:39:09 | 19 October 2012 04:22:08 | 0.44 | 0.45 |
| 29 October 2012 04:39:09 | 30 October 2012 04:22:08 | −0.13 | 0.41 |
| 10 March 2013 04:39:06 | 11 March 2013 04:22:05 | −0.49 | 0.93 |
| 21 March 2013 04:39:06 | 22 March 2013 04:22:06 | −0.17 | 0.23 |
| 12 April 2013 04:39:07 | 13 April 2013 04:22:06 | 0.39 | 0.41 |
| 23 April 2013 04:39:07 | 24 April 2013 04:22:07 | 0.00 | 0.47 |
| 6 June 2013 04:39:10 | 7 June 2013 04:22:09 | 0.31 | 0.24 |
| 17 June 2013 04:39:11 | 18 June 2013 04:22:10 | −0.13 | 0.33 |
| 28 June 2013 04:39:11 | 29 June 2013 04:22:11 | 0.39 | 0.40 |
| 9 July 2013 04:39:12 | 10 July 2013 04:22:11 | 0.10 | 0.26 |
| 11 August 2013 04:39:15 | 12 August 2013 04:22:14 | 0.73 | 0.48 |
| 13 September 2013 04:39:16 | 14 September 2013 04:22:15 | 0.10 | 0.34 |
| 29 November 2013 04:39:15 | 30 November 2013 04:22:15 | 0.06 | 0.27 |
| 10 December 2013 04:39:15 | 11 December 2013 04:22:14 | 0.16 | 0.26 |
| 21 December 2013 04:39:14 | 22 December 2013 04:22:13 | 0.46 | 0.32 |
| 30 March 2014 04:39:12 | 31 March 2014 04:22:12 | 0.57 | 0.29 |
| 10 April 2014 04:39:13 | 11 April 2014 04:22:12 | −0.25 | 0.34 |
| 29 July 2014 04:39:18 | 30 July 2014 04:22:17 | −0.00 | 0.30 |
| 9 August 2014 04:39:18 | 10 August 2014 04:22:17 | −0.03 | 0.48 |
| 19 April 2015 04:39:17 | 20 April 2015 04:22:17 | −0.31 | 0.36 |
| 22 May 2015 04:39:19 | 23 May 2015 04:22:18 | −0.53 | 0.28 |
| 2 June 2015 04:39:20 | 3 June 2015 04:22:19 | −0.52 | 0.28 |
| 13 June 2015 04:39:20 | 14 June 2015 04:22:20 | −0.23 | 0.25 |
| 7 August 2015 04:39:22 | | | |
| 9 September 2015 04:39:24 | 10 September 2015 04:22:23 | −0.86 | 0.39 |
| 20 September 2015 04:39:24 | 21 September 2015 04:22:23 | −1.46 | 0.71 |
| 1 October 2015 04:39:25 | | | |
| 3 November 2015 04:39:25 | 4 November 2015 04:22:24 | −0.12 | 0.33 |
| 14 November 2015 04:39:25 | 15 November 2015 04:22:24 | 1.20 | 0.61 |
| 28 December 2015 04:39:24 | 29 December 2015 04:22:23 | 0.12 | 0.37 |

https://doi.org/10.5194/tc-15-1-2021                                                                                          The Cryosphere, 15, 1–12, 2021

| South scene | North scene | Overlap difference Mean (m) | SD (m) |
|---|---|---|---|
| 30 January 2016 04:39:22 | | | |
| 10 February 2016 04:39:22 | 11 February 2016 04:22:21 | 0.30 | 0.49 |
| 21 February 2016 04:39:22 | | | |
| 25 March 2016 04:39:24 | 26 March 2016 04:22:23 | −0.40 | 0.42 |
| 5 April 2016 04:39:24 | 6 April 2016 04:22:23 | −0.59 | 0.51 |
| 21 June 2016 04:39:27 | 22 June 2016 04:22:27 | −1.29 | 1.08 |
| 13 July 2016 04:39:28 | | | |
| 15 August 2016 04:39:30 | | | |
| 26 August 2016 04:39:31 | | | |
| 17 September 2016 04:39:32 | | | |
| 28 September 2016 04:39:33 | | | |
| 31 October 2016 04:39:34 | | | |
| 11 November 2016 04:39:34 | | | |
| 14 December 2016 04:39:32 | | | |
| 27 January 2017 04:39:30 | 28 January 2017 04:22:29 | −0.38 | 0.40 |
| 1 March 2017 04:39:30 | 2 March 2017 04:22:29 | −0.45 | 0.30 |
| 14 April 2017 04:39:32 | 15 April 2017 04:22:32 | −0.75 | 0.43 |
| 25 April 2017 04:39:32 | | | |
| 8 June 2017 04:39:34 | 1 July 2017 04:22:35 | −1.32 | 0.75 |
| 11 July 2017 04:39:36 | 12 July 2017 04:22:35 | −1.32 | 0.64 |
| 22 July 2017 04:39:37 | 23 July 2017 04:22:36 | −0.73 | 0.46 |
| 24 August 2017 04:39:38 | | | |
| 4 September 2017 04:39:38 | 5 September 2017 04:22:38 | −1.19 | 0.59 |
| 1 April 2018 04:39:39 | 2 April 2018 04:22:39 | −0.35 | 0.33 |
| 4 May 2018 04:39:41 | | | |
| 15 May 2018 04:39:41 | | | |
| 28 June 2018 04:39:43 | | | |
| 9 July 2018 04:39:42 | 10 July 2018 04:22:42 | 0.05 | 0.31 |
| 11 August 2018 04:39:45 | | | |
| 22 August 2018 04:39:45 | | | |
| 24 September 2018 04:39:46 | | | |
| 5 October 2018 04:39:47 | 17 October 2018 04:22:47 | 0.04 | 0.77 |
| **18 November 2018 04:39:48** | 19 November 2018 04:22:47 | −0.37 | 0.33 |
| 14 February 2019 04:39:45 | 15 February 2019 04:22:44 | −0.10 | 0.27 |
| **10 April 2019 04:39:46** | | | |
| 21 April 2019 04:39:47 | 22 April 2019 04:22:46 | −0.89 | 0.36 |
| **2 May 2019 04:39:47** | **3 May 2019 04:22:47** | −0.91 | 0.40 |
| **15 June 2019 04:39:50** | 16 June 2019 04:22:49 | −0.61 | 0.32 |
| 3 October 2019 04:39:55 | | | |
| **14 October 2019 04:39:56** | | | |
| 27 November 2019 04:39:56 | 17 November 2019 04:22:55 | 0.20 | 0.40 |
| **8 December 2019 04:39:55** | | | |
| 21 January 2020 04:39:53 | 22 January 2020 04:22:53 | −0.79 | 0.36 |
| 23 February 2020 04:39:52 | | | |
| 2 November 2020 04:40:04 | 3 November 2020 04:23:03 | −0.58 | 0.44 |

*Data availability.* The NERC Polar Data Centre hosts the flow-line elevation profiles (https://doi.org/10.5285/EDE3520B-CF1C-4979-AFCC-94AC266BB61A, Bevan et al., 2021a), the elevation point time series (https://doi.org/10.5285/21B3D4FA-0EDF-4B05-B762-B4633616B0BC, Bevan et al., 2021b), the speed point time series (https://doi.org/10.5285/C0C1050A-2360-4464-9B0F-C2C101E5D1C2, Bevan et al., 2021c), and GeoTIFFs of ice surface elevation change (https://doi.org/10.5285/DF8C4AC0-1723-43AE-AD48-D02D58699F32, Bevan et al., 2021d) and ice surface speed change (https://doi.org/10.5285/668BF042-D0DE-4741-A62E-2AE93B6F7106, Bevan et al., 2021e).

*Video supplement.* Supplementary animation is available at https://doi.org/10.5285/C2DFC6B7-DD61-41F9-8624-A45F2CF978DF.

*Author contributions.* SLB created and analysed the TanDEM-X DEMs and drafted the manuscript. DIB and AJL are principal and co-principal investigators on the CALISMO project, AJL produced the Sentinel-1 velocity data, and SA contributed the ICESat-2 data. All co-authors contributed to discussions on the text.

*Competing interests.* The authors declare that they have no conflicts of interest.

*Acknowledgements.* The research was completed under Natural Environment Research Council (NERC) projects CALISMO (Calving laws for ice-sheet models) and DOMINOS (Disintegration of marine ice-sheets using novel optimised solutions). The DOMINOS project is a component of the International Thwaites Glacier Collaboration (ITGC) supported by National Science Foundation (NSF: Grant PLR 1738896). TanDEM-X data used for generating the DEMs surface velocities were supplied by DLR. Sentinel-1 data were supplied by the European Space Agency. The bed data were downloaded from the UK Polar Data Centre (https://ramadda.data.bas.ac.uk/repository/entry/show?entryid=7803de8b-8a74-466b-888e-e8c737bf21ce, last access: 5 July 2021). Many thanks to Pietro Milillo (Jet Propulsion Laboratory, California Institute of Technology, USA) for supplying the 2016/2017 grounding line locations. CE3

*Financial support.* This research has been supported by the Natural Environment Research Council (grant nos. NE/P011365/1 and NE/S006605/1).

*Review statement.* This paper was edited by Ginny Catania and reviewed by two anonymous referees.

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

**Remarks from the language copy-editor**

CE1    Gamma should only be capitalized if it is part of an official proper name. If that is not the case here, then the change will not be inserted.

CE2    Please give an explanation of why this needs to be changed. We have to ask the handling editor for approval. Thanks.

CE3    Please note the slight edit for grammatical correctness.