# Peer review of "Brief Communication: Thwaites Glacier cavity evolution"

_The Cryosphere, 2021_

## Referee Comment (RC1)

This paper provides a short communication presenting velocity and surface elevation changes over Thwaites glacier in Antarctica. Despite the presented data range between 2012-2020, new results are related to the period 2017-2020 only. The main finding of the paper is related to the underwater cavity found in previous studies at the main trunk of Thwaites glacier. The authors claim that melt at the cavity stopped during the period 2017-2020. However, these findings are only based on InSAR surface elevation changes which is typically 10 times less accurate than altimeter based surface elevation measurements.  DEM only based results can be misleading when adopted for calculating melt rates and grounding line retreats without supporting data such as DInSAR or RADAR sounders.

The authors also do not explain the methodology used for calculating melt-rates (as an example in figure 2 ice bottom crosses the bedrock). This lack makes it hard to evaluate the results. Some statements seem to be overly speculative (see comments below) and are not proved or confirmed using independent and complementary measurements.

These issues reduce my overall confidence in the main results. The presentation is also unclear at times. I suggest major revisions are required for readers to have confidence in the reliability of results and the conclusions that are presented.

Line 45 – "We tied the adjacent DEMs to their cotemporal neighbours using a point within the scene overlap. By choosing this point over a relatively flat region, height errors resulting from geolocation errors were minimised"

It is not clear to me why the authors need to coregister adjacent DEMs if they use IceSAT for calibrating the single tile. Moreover this approach, if not performed correctly might lead to errors related to geocoding or mi-registration (see schematics below). Hence a single tie point seems to be a weak approach for tackling this problem.

[Figure]

[Figure]

54-55 "Small tidal Ranges" of 0.5 meters would correspond to abut 5 meters of difference when considering multiplying by the flotation factor. This factor should be taken into account if you are looking at grounding zones or a more solid justification of why this was not taken into account should be provided.

65 Why do you need tie points described on line 45 if you can calibrate data with icesat?

69 TerraSAR-X SLCs (2012–2014) why only this period and not the entire dataset? Moreover, you can do pixel tracking with DEMs itself if pairs far apart in time are characterized by Low SNR levels in the Pixel Offset maps.

73 What does sampling every 100 m means? Do you pick a cross-correlation window every 100 m (for Sentinel) and 40 m for (TSX)? What are  other parameters like Window size and widow search used for different sensors? Depending on the used parameters the velocity maps will have different accuracies

79 It does not look like there is good agreement betweem the DEM based results compared to the GL west the grounding line (Fig 1.a). The TDX backscatter intensity covers the elevation of floating areas. How the DEM derived grounding lines have been obtained has not been explained anywhere in the manuscript.  This part requires an extensive rewriting.

85-89 This statement is not supported by the data since the DEM based Grounding lines are 2 orders of magnitude less sensitive compared to the InSAR based Grounding lines. This also is shown in Figure 1a West of thwaites main trunk where no grounding line is detected by the "DEM based" grounding line.  As explained in Milillo et al 2019 a grounding line retreating could show up as a surface uplift simply because bending forces at the grounding line get released once the grounding line retreats. This effect would alter height above flotation calculations together with the fact that the grounding line is not in hydrostatic equilibrium. Since you are not comparing Grounding lines obtained with the same measurements this statement seems too speculative.

90-92 This sentence is not supported by any data in the manuscript. How do you identify the cavities? This seems to be part of the discussion and not of the results.

109 how do you calculate the cavity volume? This is not explained in the manuscript

129-130 This sentence is not supported by observations or data and seems overly speculative.

Figure 1a Instead of surface elevation please show height above flotation since you are comparing grounding line measurements. Add legend with Grounding line acquisition dates

Figure 2 Despite the nice colorbar It seems counterintuitive to look at the bottom of the ice extending below the bedrock level. Since no methodology for retrieving these thickness change measurements has been described it is really hard to evaluate this figure and comment on it.

---

## Author Comment (AC1)

**INTRODUCTION:** We thank Reviewer 1 for their insightful comments. In some cases these seem to arise because of misunderstandings of what we have done or how we have done it. In these cases, we will review and make additions to the text to further clarify our methods and findings. In other cases, where the comments were justified, we will make changes to the way we carry out the analysis to improve the research and the way it is presented. Details of changes proposed are given which replies to each comment below.

*Reviewer: Despite the presented data range between 2012-2020, new results are related to the period 2017-2020 only.*

**Reply:** We present independently-derived data for the entire period to validate previous work and to provide more detail and insight into those findings, as well as new data for the more recent period which is the main focus of the paper. We chose *Brief Communications* for publication because our aim is to provide an update on previous work. WE THEREFORE PROPOSE NO CHANGES.

*Reviewer: The main finding of the paper is related to the underwater cavity found in previous studies at the main trunk of Thwaites glacier. The authors claim that melt at the cavity stopped during the period 2017-2020.*

**Reply:** Nowhere in the paper do we claim that melt at the cavity stopped in 2017. We simply show that the cavity stopped growing in response to ongoing melt at the grounding line, and that this is consistent with models. WE THEREFORE PROPOSE NO CHANGES.

*Reviewer: However, these findings are only based on InSAR surface elevation changes which is typically 10 times less accurate than altimeter based surface elevation measurements.*

**Reply:** Using DEMs is a highly complementary approach to DInSAR for determining the boundary between floating and grounded ice. Although less precise than altimeter measurements, the advantage of InSAR elevations is the repeat and complete spatial coverage. InSAR DEMs can be as accurate as the data used to tie them vertically, which in our case is altimetry. Far from being misleading, our results confirm those of previous studies and go on to provide further insights into processes and recent evolution. WE THEREFORE PROPOSE NO CHANGES.

*Reviewer: DEM only based results can be misleading when adopted for calculating melt rates and grounding line retreats without supporting data such as DInSAR or RADAR sounders.*

**Reply:** The reviewer's claim that we do not use supporting data is incorrect, and in any case we do not calculate melt rates but simply report on cavity geometry. We validate our elevations during the period when both InSAR and altimetry data are available and find a mean and standard deviation of the difference (TDX – IS2) to be -0.7 m and 2.2 m, respectively, over 23,000 points in the south scene, and +0.2 m and 1.8 m over 877 points in the north scene. This is a level of correspondence which is more than sufficient to support

our conclusions. WE WILL MAKE ADDITIONS TO THE TEXT TO HIGHLIGHT THIS EXCELENT CORRESPONDENCE.

*Reviewer: The authors also do not explain the methodology used for calculating melt-rates (as an example in figure 2 ice bottom crosses the bedrock). This lack makes it hard to evaluate the results.*

**Reply:** We do not claim to calculate melt rates and the method for inferring thickness and flotation from surface and bed heights is explained in full in lines 50-62. The crossing of ice-base and bedrock lines in Figure 2 results from the chosen method of presentation which focusses on height above flotation. CHANGES PROPOSED: To clarify our approach to presenting the results in this way we will include the following explanation in the revised text: Where the ice is grounded the plots show the ice base below bedrock, this is simply a device to indicate height above flotation scaled by $\rho w/(\rho w - \rho i) \approx 8$.

*Reviewer: Some statements seem to be overly speculative (see comments below) and are not proved or confirmed using independent and complementary measurements. These issues reduce my overall confidence in the main results.*

**Reply:** Wherever possible we have compared our results directly to independent measurements, including our grounding lines against published DInSAR ones. The figures show very clearly an excellent correspondence between our estimate of the grounding lines and those produced using DInSAR. We also compare our surface elevations with ICESat-2 elevations. We therefore find the assertion that our data are not confirmed by complementary measurements to be unsubstantiated. WE THEREFORE PROPOSE NO CHANGES.

*Reviewer: The presentation is also unclear at times. I suggest major revisions are required for readers to have confidence in the reliability of results and the conclusions that are presented.*

**Reply:** This comment is not sufficiently specific to allow us to respond. WE THEREFORE PROPOSE NO CHANGES TO THIS SPECIFIC COMMENT, ALTHOUGH CHANGES ELSEWHERE SHOULD CLARIFY THE PRESENTATION.

*Reviewer: Line 45 – "We tied the adjacent DEMs to their cotemporal neighbours using a point within the scene overlap. By choosing this point over a relatively flat region, height errors resulting from geolocation errors were minimised". It is not clear to me why the authors need to coregister adjacent DEMs if they use IceSAT for calibrating the single tile. Moreover this approach, if not performed correctly might lead to errors related to geocoding or mi-registration (see schematics in the PDF version). Hence a single tie point seems to be a weak approach for tackling this problem.*

**Reply:** We took the only reasonable approach to vertically calibrating the DEMs based on the availability of ICESat-2 data, which did not begin operation until October 2018. It was necessary to use the landward DEM frame as a source of reference for the seaward frame

because ICESat-2 data is sparse and unreliable in the latter. Nevertheless, since our findings are based predominantly on the landward frame, error propagation from frame to frame has no bearing on the main result. However, we agree that this approach needs to be explained better and the error arising from it could be better presented. CHANGES PROPOSED: We will list in Table A1 the mean elevation differences between adjacent scenes in a 4 x 4 km (500 x 500 pixels) patch above the grounding line and in the overlap region. The majority of the mean and standard deviations of the differences are less than 1 m and therefore well within the error margin for our approach.

***Reviewer:*** *54-55 "Small tidal Ranges" of 0.5 meters would correspond to about 5 meters of difference when considering multiplying by the flotation factor. This factor should be taken into account if you are looking at grounding zones or a more solid justification of why this was not taken into account should be provided.*

**Reply:** This is a valid point. CHANGES PROPOSED: We will extract tidal heights from a point close to the cavity and use these to correct height above flotation as presented in Figs. 1 and 2, and a new Appendix Figure A1. Preliminary analysis shows that it makes little visible difference to grounding line location. However, the work will be more valid for having done this.

***Reviewer:*** *65 Why do you need tie points described on line 45 if you can calibrate data with Icesat?*

**Reply:** Reponses to previous comments will have addressed this issue.

***Reviewer:*** *69 TerraSAR-X SLCs (2012–2014) why only this period and not the entire dataset? Moreover, you can do pixel tracking with DEMs itself if pairs far apart in time are characterized by Low SNR levels in the Pixel Offset maps.*

**Reply:** Thank you for spotting this apparent omission. CHANGES PROPOSED: We did track and had plotted TerraSAR-X SLCs beyond 2014 and will now highlight these points on Fig. 3.

***Reviewer:*** *73 What does sampling every 100 m mean? Do you pick a cross-correlation window every 100 m (for Sentinel) and 40 m for (TSX)? What are other parameters like Window size and widow search used for different sensors? Depending on the used parameters the velocity maps will have different accuracies.*

**Reply:** Thanks for asking for more detail and clarification in our feature-tracking approach. CHANGES PROPOSED: We will add much more detail of the feature-tracking and filtering method. This could be within the main text or as an Appendix if it becomes too unwieldy.

***Reviewer:*** *79 It does not look like there is good agreement between the DEM based results compared to the GL west the grounding line (Fig 1.a). The TDX backscatter intensity covers the elevation of floating areas.*

**Reply:** The agreement between DInSAR grounding lines and our DEM-based grounding lines in Fig. 1a is in fact remarkably good and shows the quality that can be achieved with this technique. What we think the reviewer is referring to is the area above the cavity and the area to the west of the cavity where there appears to be some disagreement. We agree that these issues need further clarification. CHANGES PROPOSED: We have obtained updated grounding lines from Pietro Milillo which has removed the disagreement to the west of the cavity. We will also redesign Figure 1 to make everything clearer and add similar figures for other years (Appendix) for comparison. The apparent differences above the cavity are explained by the temporal changes in grounding line here, and the nature of this region being more akin to a grounding zone. We will make changes to the text to explain this and clarify in detail where there are apparent differences between the two techniques and what they mean.

*Reviewer:* *How the DEM derived grounding lines have been obtained has not been explained anywhere in the manuscript.  This part requires an extensive rewriting.*

**Reply:** The DEM regions of flotation/grounding are where the hydrostatic thickness is less/greater than the difference between surface elevation and bedrock elevation. This is explained this at line 56. CHANGES PROPOSED: We will review this text to make sure that the method is more clearly presented and flagged.

*Reviewer* *85-89 This statement is not supported by the data since the DEM based Grounding lines are 2 orders of magnitude less sensitive compared to the InSAR based Grounding lines. This also is shown in Figure 1a West of thwaites main trunk where no grounding line is detected by the "DEM based" grounding line.  As explained in Milillo et al 2019 a grounding line retreating could show up as a surface uplift simply because bending forces at the grounding line get released once the grounding line retreats. This effect would alter height above flotation calculations together with the fact that the grounding line is not in hydrostatic equilibrium. Since you are not comparing Grounding lines obtained with the same measurements this statement seems too speculative.*

**Reply:** The correspondence between DEM-based and DInSAR grounding lines everywhere except above the cavity and to the west of the cavity clearly demonstrates that the DEM-based approach (with calibration by precise laser altimetry) is entirely valid at the level of precision required for the claims we make. The discrepancy to the west of the cavity has been corrected by updated DInSAR grounding lines. The temporal evolution of the cavity (Figure 2) shows very clearly that our approach is sufficiently sensitive to changes in the cavity. CHANGES PROPOSED: We will clarify the method with: 'Using InSAR DEMs to delineate grounding lines or zones is a complementary method to using DInSAR. The DEM method requires accurate surface and bed elevations, and a good estimate of ice density whereas the DInSAR method requires a detectable response of the floating ice to changing tidal elevations. Using the DEM method has allowed us to create repeated full-coverage mapping and to identify locations where there is insufficient ice-shelf flexure to allow detection by the DInSAR method but where ocean water ingress is possible and likely to result in further melt. The DEM method may miss a grounding line retreat where the grounding line is not in hydrostatic equilibrium and retreat may even cause surface uplift. It

will be important to confirm the evolution of the grounding lines with interferometric analysis.'

*Reviewer: 90-92 This sentence is not supported by any data in the manuscript. How do you identify the cavities? This seems to be part of the discussion and not of the results.*

**Reply:** We refer to a cavity as being where the ice is assumed to be floating, that is where the surface elevation minus the hydrostatically calculated thickness is above the bedrock elevation. Our figures support this sentence, but we will review the text to make sure everything is clear. CHANGES PROPOSED: review and clarification of the text.

*Reviewer: 109 how do you calculate the cavity volume? This is not explained in the manuscript.*

**Reply:** We do not claim to calculate the cavity volume, only elevation, inferred thickness and resulting floating/grounded areas. We use changes, or lack of changes, in these to infer the evolution of the cavity. NO CHANGES PROPOSED.

*Reviewer: 129-130 This sentence is not supported by observations or data and seems overly speculative.*

**Reply:** Accepted. CHANGES PROPOSED: we will remove this sentence.

*Reviewer: Figure 1a Instead of surface elevation please show height above flotation since you are comparing grounding line measurements. Add legend with Grounding line acquisition dates.*

**Reply:** CHANGES PROPOSED: We will redraw Fig. 1a as suggested and added a legend.

*Reviewer Figure 2 Despite the nice colorbar It seems counterintuitive to look at the bottom of the ice extending below the bedrock level. Since no methodology for retrieving these thickness change measurements has been described it is really hard to evaluate this figure and comment on it.*

**Reply:** CHANGES PROPOSED: we explain how to interpret the lines crossing the bedrock in the caption to Fig. 2. We will also add an explanation to the main text. We explain how the thickness was calculated in Lines 50 to 62.

---

## Author Comment (AC2)

*Reviewer:* The presented brief communication article deals with the formation and temporal evolution of a previously identified cavity close to the grounding line at Thwaites glacier. Recent studies identified tipping points for a continued grounding line retreat for Pine Island glacier also located at WAIS [1]. Similarly Thwaites glacier has also been implicated to experience continued retreat of the grounding line [2]. Due to the fast ice flow at the main trunk of Thwaites it is currently challenging to update grounding line positions from InSAR acquisitions. It only remains possible with a small temporal baseline (COSMOS-Skymed, Milillo 2019). Therefore, a detailed time series of grounding line positions is of high interest to the scientific community as it allows to under investigate the melt processes and timescales of grounding line migration directly at the grounding lie, although a derivation of the grounding line from height above floatation is less accurate than InSAR derived grounding lines. It has to be noted that accurate bathymetry and density assumptions are crucial for correct grounding line positions. The used TDM data (if properly calibrated) is an accurate enough reference for the surface elevation. From the time series of height above floatation measurements the authors concluded that the previously reported cavity remained stable in height and extent and therefore the grounding line position also remained at a position of a slightly upward sloping bed which is predicted by coupled ice–ocean models.

**Reply:** Thank you for your thorough review.

*Reviewer:* I have some points that need further addressing before publication
- My first comment concerns the vertical calibration of the TDM time series. It is currently only explained in a few sentences. It would be preferable to use more than one IceSAT-2 measurement for calibrating the TDM scene to the IceSAT data. I suggest adding the used IceSAT-2 track in one of the Figures for a better overview. If surface elevations over the crevassed floating parts are used, it is important to calibrate the TDM data also in areas of limited or no signal penetration. A statement about the surface roughness, or distance to the area of investigation should be included. If the surface is rough and crevassed I would see no problem in selecting the area as it was done in the article but the argument is missing. Also a statement about the size of the calibration area is missing. What diameter does the footprint of the IceSat measurement have in the ATL06 data and how what size of TDM area was it compared against? Regarding the calibration of neighboring scenes in the range direction one hast to be careful to also include the TDM baseline uncertainties in the error budget of the adjacent scenes, as the two scenes are not from the same track and can be characterized by different baseline errors. A baseline uncertainty of 1mm depending on the height of ambiguity adds elevation uncertainties in the order of 1m [3]. Depending on the used method for vertical calibration to IceSAT-2 a tilt in range could be remaining and propagate to the neighboring scene.

**Reply:** We agree that it would have been ideal to calibrate every DEM with ICESat-2 data. However, ICESat-2 did not acquire data until October 2018 and then only 8 of the TanDEM-X scenes were acquired within 3 days of any ICESat-2 data within their geographic coverage. We therefore used the same point for all DEMs, chosen in a high-elevation, slow-moving location to minimise temporal elevation change at that point. CHANGES PROPOSED: We will expand this section as below and add the ICESat-2 track used to Fig. 1a.

The lack of co-temporal ICESat-2 data also explains why we were unable to calibrate the DEMs in the fast-moving floating areas. The technique for creating interferometric DEMs does not easily lend itself to spatially varying calibration although we accept that varying penetration depths may be an issue. As mentioned in lines 65 to 67 we were able to make an assessment of the vertical accuracy using the post-2018 co-temporal TanDEM-X/ICESat-2 data. The mean and standard deviation of the difference over 24,000 ICESat-2 points being -0.57 m and 2.25 m, respectively. We will produce a map of these point differences for the appendix.

The ICESat-2 footprint is 17 m and we compare the centre of this footprint to an individual 8 m TanDEM-X DEM pixel. We will add this information to the text.

The propagation of baseline errors into the adjacent scene is a good point. We will include a sentence about this although point out that the analysis is mostly confined to the scene that is directly tied to ICESat-2. Thank you also for the Rizzoli et al. (2017) reference, we will amend our estimate of baseline errors to 1 m propagating to 2 m within the neighbouring scene.

*Reviewer: The actual derived grounding line from height above floatation is not displayed in Figure 1. The caption states only MEaSUREs (purple: 1996 and yellow: 2011) and Milillo et al. (white: 2019). A time-series of 2D grounding lines would strengthen the argument of the suitability of height above floatation in this case, especially as it allows for a comparison with InSAR derived grounding lines over the whole area. L. 79 suggests that this was done. If a 2D representation of the grounding line time-series does not reproduce previous InSAR results over the entire area, it has to be stated that the analysis is restricted to the area of the cavity. In this case, results from height above floatation could be calibrated to the InSAR grounding line position.*

**Reply:** CHANGES PROPOSED: We have recreated Fig. 1a using height above flotation rather than mapping floating areas, and this makes the DEM-based grounding line much clearer. For Fig. 1a we will now use the most recent DEM (Nov 2020) but also map heights above flotation for 2011, 2014 and 2017 for direct comparison with DInSAR grounding lines as a figure for the appendix, hence addressing the request for 2D representation of the grounding line. It is not simple to match individual 2016/2017 DInSAR grounding lines with DEM ones as the dates do not match exactly and, as reported by Milillo et al. (2019), the DInSAR lines cover a grounding zone that migrates with the tides over 2.5 km.

We would also like to mention here that following communication with Pietro Milillo regarding exact dates for the 2016/2017 grounding lines we obtained new shapefiles which map the GLs differently to those we had earlier. We do not know where the discrepancy originated but are assured that the GLs now mapped are correct.

*Reviewer: The discussion and especially the link to coupled models L 115-120 is difficult to understand. For me the physical process of why a stable grounding and cavity volume is reached after several years (how many?) is not entirely clear. Is this predicted by these models because they take ocean circulation of warm water in the cavities into account? If other models are used, would they predict a growing cavity and subsequent grounding line retreat (L. 120)? How the increasing velocity Fig 3, A1 are used in the arguments from L. 124-134 is not clear. I do not understand the meaning of this sentence "However, bed*

*topography and ice-thickness close to floatation can superimpose rapid local change on the background long-term evolution of Thwaites and other WAIS glaciers in ASE"*

**Reply:** Indeed, the coupled models are able to take into account the response of ocean circulation to changing geometry. Yes, without coupled modelling grounding lines continue to retreat as we do say in Line 20. CHANGES PROPOSED: In terms of time we will be more precise and change 'a couple' to one or two as per Goldberg et al. (2012).

At line 26 we are saying that long-term steady acceleration and thinning are taking place on grounded ice but that locally, where ice is close to flotation, thinning can induce flotation, allow ocean ingress, and rapidly induce a melt/thinning feedback until ocean circulation adjusts.

We will rephrase as 'Bed topography and ice-thickness close to flotation can mean that the observed long-term and steady thinning of Thwaites and other WAIS glaciers in ASE can cause a rapid local thinning induced by a melt/flotation feedback, until ocean circulation adjusts.'

*Reviewer: Overall the article is of high scientific interest and well presented with clear language. The raised concerns require major revisions.*

*[1] Rosier, Sebastian H. R., Ronja Reese, Jonathan F. Donges, Jan De Rydt, G. Hilmar Gudmundsson, und Ricarda Winkelmann. 2021. „The Tipping Points and Early Warning Indicators for Pine Island Glacier, West Antarctica". _The Cryosphere_ 15 (3): 1501–16. [https://doi.org/10/gjnwhg](https://doi.org/10/gjnwhg).*

*[2] Joughin, I., Smith, B. E., and Medley, B.: Marine ice sheet collapse potentially under way for the Thwaites Glacier basin, West Antarctica, Science, 344, 735–738, [https://doi.org/10.1126/science.1249055](https://doi.org/10.1126/science.1249055), 2014.*

*[3] Rizzoli, Paola, Michele Martone, Carolina Gonzalez, Christopher Wecklich, Daniela Borla Tridon, Benjamin Bräutigam, Markus Bachmann, u. a. 2017. „Generation and Performance Assessment of the Global TanDEM-X Digital Elevation Model". _ISPRS Journal of Photogrammetry and Remote Sensing_ 132 (Oktober): 119–39. [https://doi.org/10.1016/j.isprsjprs.2017.08.008](https://doi.org/10.1016/j.isprsjprs.2017.08.008).*

**Reply:** Thanks for your help in making these revisions easy to implement.

*Reviewer: - L. 4 continued*

**Reply:** Corrected.

*Reviewer: - L. 19 mention the used data ERS, COSMO-Skymed for deriving the grounding lines*

**Reply:** Will do.

*Reviewer: Choosing only one tie point is not robust*

**Reply:** We have explained above the necessity for this approach, and we have further also used ICESat-2 data to validate the DEM heights to demonstrate the robustness of this approach.

*Reviewer: Combining adjacent across trade scenes in the overlap region includes baseline errors of in the order of 1m*

**Reply:** A new presentation of the error budget and its implications will include this point.

*Reviewer:    - Depending on the surface properties of chosen point there might be an elevation bias due to signal penetration.*

**Reply:** This will be covered in the new discussion of errors.

*Reviewer:    - Is the same point IceSAT-2 measurement used for the entire time series? If so, thinning rates should be close to 0 and quantified at this location from an independent source.*

**Reply:** This will be covered in the new discussion of errors. We will also refer to this in discussing long-term elevation change shown in Fig. 3.

*Reviewer:    - Show IceSat 2 track on Fig 1*

**Reply:** Will do.

*Reviewer: - L. 54 What is the result of 0.5m tidal variation in thickness change?*

**Reply:** In response to comments from Reviewer: 1 we will include a tidal correction.

*Reviewer: - L. 68. Combining errors: baseline, tidal range, TDM orbit. height above floatation from the two scenes will be characterized by different errors*

**Reply:** Agreed and we will add a more thorough discussion as shown above.

*Reviewer: - L. 78 Could you calibrate f_a on the previous InSAR grounding line positions*

**Reply:** This could be an interesting exercise. A problem would be that DInSAR grounding line locations are very tide dependent and we are unlikely to have a TanDEM-X DEM coinciding exactly in time and tide with the DInSAR mapping.

*Reviewer: - L. 80 loss of 50 to 60m*

**Reply:** Will change, thank you.

*Reviewer: - L. 88 Quantify value. How many meters above flotation is reported. Could this be explained by erroneous bathymetry?*

**Reply:** Now that we are using revised grounding line vectors from Milillo, this discrepancy has been removed.

*Reviewer: - L. 97 good agreement to InSAR grounding line locations*

**Reply:** Will change.

*Reviewer: - L. 112 thick → deep - what does imprinted with bed topography mean?*

**Reply:** We will change thick to deep. Jordan et al. (2020) were observing that the ice shelf base still contained a signature of the bed implying that melt rates were low. We will change the sentence to:

'and the sub-shelf topography continues to closely follow the contours of the bed topography

*Reviewer: - L. 130 Not justified - Discussion is hard to follow*

**Reply:** As requested also by Reviewer: 1 we will remove this sentence.

*Reviewer: - Fig. 1     - Show IceSAT-2 tracks*

**Reply:** Will do.

*Reviewer:     - Missing 2D time series of height above floatation derived grounding line positions*

**Reply:** We will map 2020 in Fig. 1a, and include three (2011, 2014 and 2017) maps in the Appendix.

*Reviewer:     - A legend would be helpful. MEaSUREs (purple, yellow), Milillo et al. (white)*

**Reply:** Yes, will do.

*Reviewer:     - Fig2: Reword caption: Surface elevation and basal elevation inferred from hydrostatic thickness. The thickness itself is not plotted.*

**Reply:** Good point, will do so.

*Reviewer:     - Can you also quantify the scaling factor as it was used in the study here.*

**Reply:** Scaling factor is approximately 8 for these ice thicknesses. We will include this information in the caption.

*Reviewer:* - *I cannot distinguish the colors of the arrows. The arrows should be labelled.*

**Reply:** Good point, will do so.